# Generation of swine movement network and analysis of efficient mitigation strategies for African swine fever virus

**Tanvir Ferdousi**[1]*, **Sifat Afroj Moon**[1], **Adrian Self**[2], **Caterina Scoglio**[1]

**1** Department of Electrical and Computer Engineering, Kansas State University, Manhattan, Kansas, United States of America, **2** National Agricultural Biosecurity Center, Kansas State University, Manhattan, Kansas, United States of America

* tanvirf@ksu.edu

## Abstract

Animal movement networks are essential in understanding and containing the spread of infectious diseases in farming industries. Due to its confidential nature, movement data for the US swine farming population is not readily available. Hence, we propose a method to generate such networks from limited data available in the public domain. As a potentially devastating candidate, we simulate the spread of African swine fever virus (ASFV) in our generated network and analyze how the network structure affects the disease spread. We find that high in-degree farm operations (i.e., markets) play critical roles in the disease spread. We also find that high in-degree based targeted isolation and hypothetical vaccinations are more effective for disease control compared to other centrality-based mitigation strategies. The generated networks can be made more robust by validation with more data whenever more movement data will be available.

**Data Availability Statement:** All relevant data are within the manuscript and its Supporting Information files.

## Introduction

Animal movement networks are important to model disease outbreaks and identify the pathways of disease spread. In the US, pig farm data including herd sizes, geolocations, and movements between farms are difficult to obtain due to the sensitive nature of data and potential economic risk of making such information public. Epidemiologists and other researchers who need such data have to rely on models that can disaggregate available county or state level data. One such example is the work of Burdett et al., who developed a simulation model to quantify pig population and generate geolocation of individual farms [1]. However, this model does not produce movement data. In another work by Valdes-Donoso et al., machine learning techniques were used to predict movement networks in the State of Minnesota [2]. A recent work uses a maximum information entropy approach to estimate movement probabilities among swine farms [3] and suggests that the 'small-world phenomenon' could make the US swine industry vulnerable to infectious disease outbreaks. Despite several efforts, pig level networks in the US swine industry are not readily available for simulating disease outbreaks. One way to

**Funding:** TF & SAM: Supported by the NSF\NIH\USDA\BBSRC Ecology and Evolution of Infectious Diseases (EEID) Program through USDA-NIFA Award 2015-67013-23818 and by the State of Kansas, National Bio and Agro-Defense Facility (NBAF) Transition Fund through the National Agricultural Biosecurity Center (NABC) at Kansas State University. The funders had no role in study design, data collection and analysis, decision to publish, or preparation of the manuscript.

**Competing interests:** The authors have declared that no competing interests exist.

overcome this issue is to design a network generator that can produce synthetic swine networks given some of the available movement network characteristics and census data.

There has been substantial work in the area of graph generation. The most basic random graph model is the Erdös—Rényi model [4] that can produce graphs with a certain edge probability between any pair of vertices. The vertex degrees of such random graphs follow the Poisson distribution [5]. There are several mechanisms to generate graphs with prescribed degree sequences. Milo et al. describes [6] two mechanisms: switching algorithm [7, 8] and matching algorithm [5, 9]. In the switching algorithm, graphs are generated based on a degree sequence and the edges are shuffled without changing the degrees to introduce randomness. The matching algorithm is also called the configuration model [10] where stubs (open ended handles) are assigned to vertices and later joined pairwise completely at random. Our limited movement data situation with a swine movement network presents us with a unique challenge where we have several different vertex types with their given average in/out degrees and their range of values [2]. We also have the probability of having a directed edge from one vertex type to another. Using these two sets of data, we design a network generator that uses a modified version of the configuration model and the generalized random graph model [10]. Generated random graphs have been used for various purposes that includes running outbreak simulations [11] and predicting the impacts of disease control [12]. Pig movement networks have been analyzed and found to be useful in predicting the risk of infectious disease outbreaks [13]. The effects of immunizations based on network centrality metrics have been explored before [14, 15] for human diseases and such studies can suggest efficient strategies for disease control. In this paper, we use several proven network metrics to understand disease spreading phenomena in pig networks.

African swine fever (ASF) is a highly contagious infection that poses as a threat for the pork industry due to its high mortality and no effective vaccine or cure [16]. Several recent outbreaks in Romania, Bulgaria and Belgium have already threatened European pork producers [17, 18]. China, the largest pork producing country has an ongoing ASF outbreak and has reportedly culled 1,170,000 hogs as of 3[rd] October 2019 [19]. They reported their first outbreak in early August 2018 and since then there have been about 158 outbreaks in 32 provinces [19]. Several major Chinese pork producers have cut their profit forecasts, some of them are expecting as much as 80% reduction compared to 2017 [20]. The Chinese officials have undertaken several methods in order to control the outbreaks that include, culling of all pigs within 3km of the infected area, pig movement restrictions, surveillance around containment/protection zones, and destruction of pig products [21]. The analysis of Herrera-Ibata et al. finds that although US has a low risk of ASF introduction overall, multiple states such as Iowa, Minnesota, and Wisconsin are the ones to be more vigilant about for an ASF introduction by the legal import of live pigs [22]. There have been several attempts to model ASF outbreaks. Barongo et al., used a stochastic compartmental model to investigate the effects of control measures on ASFV and found that early intervention can help in managing the ASF epidemics [23]. The effects of residue from deceased animals were included in the work of Halasa et al. to simulate the spread of ASFV [24]. Using transmission experiments on the Georgia 2007/1 ASFV strain, Guinat et al. estimated pig-to-pig transmission parameters for both within pen and between pen infections and they found the reproductive ratios to be 5.0 and 2.7 respectively [25]. On the other hand, Gulenkin et al. estimated the basic reproductive ratio for the outbreaks in the Russian Federation to be 8-11 within the infected farms and 2-3 between farms [26]. Barongo et al. also estimated this ratio for Uganda outbreaks to be in the range of 1.58-3.24 depending on various estimation methods they used [27]. In another work, Guinat et al. inferred transmission parameters using pig mortality data [28]. A recent work by Hu et al. used Bayesian inference on previous transmission experiments [25] to account for

unobserved infection times and latent periods [29]. Most of the ASFV research is focused on parameter estimates while several others investigate virus importation risk in US mainland. Despite the numerous studies, there is a lack of knowledge on how the swine industry in the US would be affected in case an ASFV outbreak starts in the US.

The contributions of this paper are several: i) we propose a swine movement network generator, ii) we run ASFV epidemic simulations and compare how different farm operation types affect the outbreak dynamics, and iii) we analyze and compare the effectiveness of multiple centrality based targeted control measures. In the *Results* section, we describe our generated farm level network along with the outcomes of preliminary network analyses. We also explain the ASFV outbreak simulation results and compare different operation types as sources of infection. Finally, we investigate the impact of different disease control strategies. The *Materials and Methods* section contains detailed information on swine movement data, network generation, analysis methods, ASFV epidemic model, and its parameters. The pseudocodes for the algorithms are detailed in S1 Appendix.

## Results

### Movement network

The generated farm level movement network is shown in Fig 1. This directed network contains 84 farms from two Minnesota counties (Stevens and Rice). There are five different swine operations marked as: Boar Stud (B), Farrow (F), Nursery (N), Grower (G), and Market (M) with 3, 22, 12, 39, and 8 sites respectively. A visual inspection of Fig 1 suggests that the movement of pigs start from farrow and nursery operations and end at the markets while a large number of grower farms lie in those paths. We also analyze the node centrality measures of the generated network which are shown in Fig 2. As the network is generated based on degree centrality data (Table 2), it is expected that the results shown in this figure ($K_{in}$ and $K_{out}$) would resemble it. The market operations have significantly high in-degree centralities (median value of 9) while the nursery operations have high out-degree centralities (median value of 3) followed by farrow and grower operations (both with median values of 2). The farrow operations have high betweenness values (median of 8.9167) followed by grower operations (median of 4).

To understand how the connectivity in the farm network can be disrupted, we perform a robustness analysis. Based on the node centrality measures of the network, we rank the nodes in a decreasing order and create three lists ($K_{in}$, $K_{out}$, and $BC$). Going through those lists, we remove (isolate) nodes one by one from the network and compute the largest connected component in every step. The results are depicted in Fig 3, where the relative sizes of the largest components are plotted against three centrality based node removal/isolation schemes. While all three schemes decrease the component sizes, the removal of high $K_{in}$ nodes demonstrates relatively better outcome in breaking the network. Approximately 94.1% of the farms in total can be isolated from the original network by isolating only 33.3% of the high in-degree farm nodes. For the other two schemes, isolation of 33.3% high centrality ($BC$ and $K_{out}$) farms will isolate about 38.1% of the farms in total. The in-degree centrality based isolation strategy shows a significant (about $\sim$ 2.5 times) improvement over other options.

### Outbreak dynamics

In a generated swine pig level network of the two Minnesota counties, we introduce an ASFV outbreak by choosing a pig farm uniformly at random as the seed farm. Within this selected farm, we infect at most 10 (if there are more than 10) pigs to introduce the pathogen and observe the progression of the disease spread. The averaged out results of 1000 independent simulations are shown in Fig 4. We use the parameter values given in Table 5. For the infection

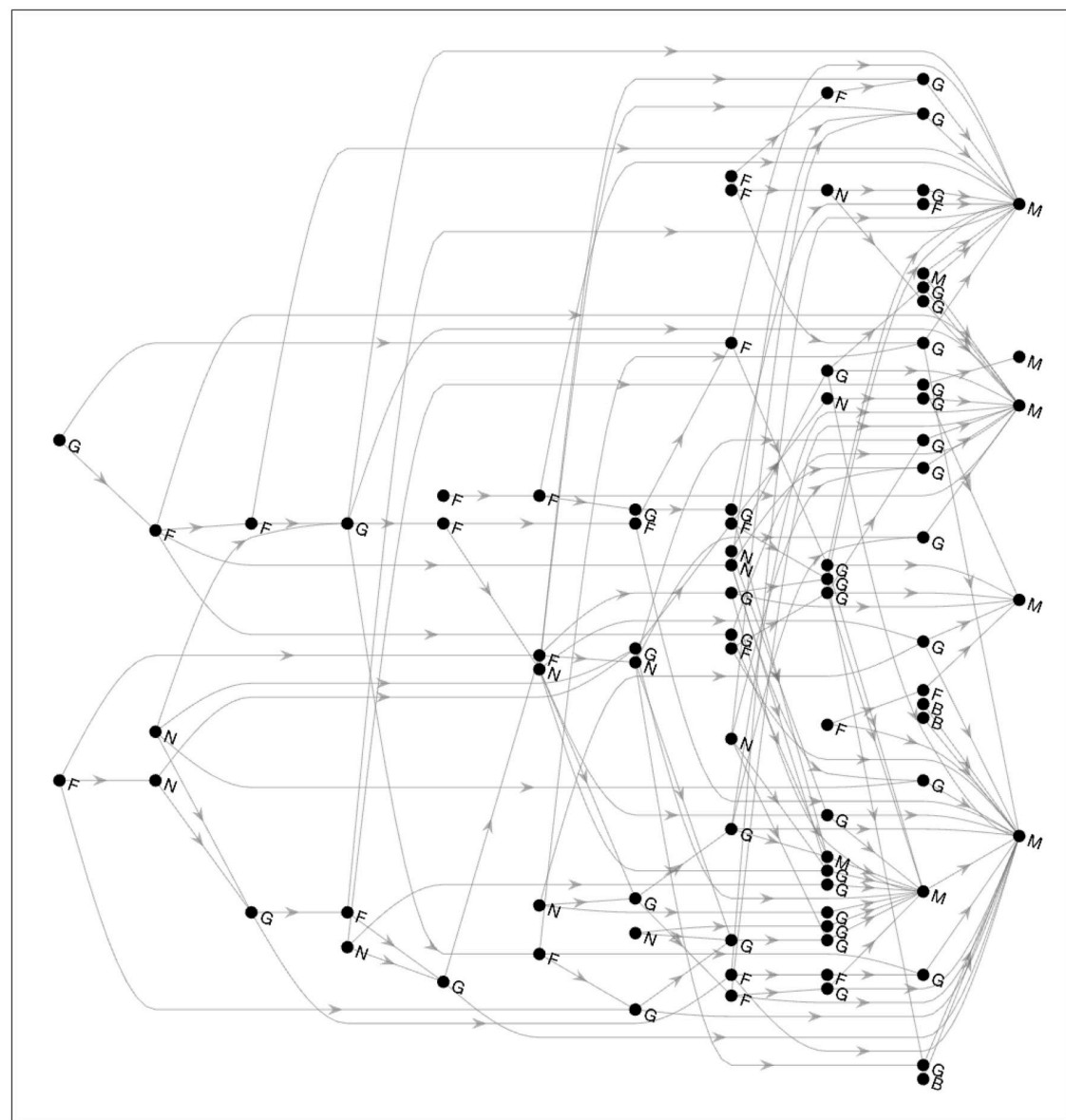

**Fig 1. Generated farm level swine movement network.** The graph shows the generated network at the farm level. The solid circles (nodes) indicate swine operations and the gray arrows connecting them indicate pig shipments with directions. The swine operations (nodes) are labeled according to their types: Boar Stud (B), Farrow (F), Nursery (N), Grower/Finisher (G), and Market/ Slaughterhouse (M).

rate, $\beta$, we use the median value given in Table 5 along with the values 25% above and below the median as indicated in the legends of the plots in Fig 4. We observe outbreaks lasting about 378 days for the median value of $\beta$ which infects about 1.84% [95% CI 1.65 2.03] of the pork population. For a network of 249,150 pigs, this roughly translates to about 4,584 [95% CI 4,111 5,047] pigs dying from the outbreak. A 25% increase in $\beta$ would lengthen the outbreak duration by about 33% and affect twice as many pigs. A 25% reduction in $\beta$ shortens the outbreaks by 32% and reduces the outbreak size by 59.8%. For the $\beta$ value around the median and above, the outbreak reaches its peak within 95-100 days and for the $\beta$ values below the median, the outbreaks do not surpass the initial fraction of infected pigs.

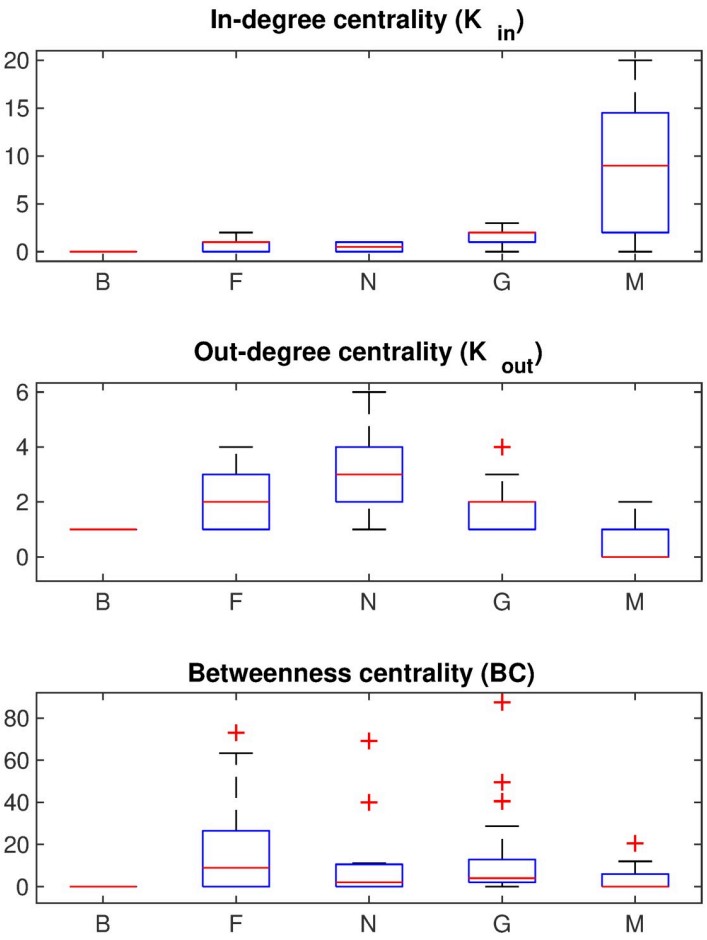

**Fig 2. Centrality measures of the generated network.** The three set of boxplots show three different centrality measures as marked (In-degree ($K_{in}$), Out-degree ($K_{out}$), and Betweenness ($BC$)). The five different pig operations are marked in the horizontal axes as: Boar Stud (B), Farrow (F), Nursery (N), Grower/Finisher (G), and Market/ Slaughterhouse (M). Each boxplot shows the range between $25^{th}$ and the $75^{th}$ percentiles (blue box) and the median (red line). The values outside 1.5 times the inter-quartile range are marked as outliers (+ signs).

For the results of Fig 4, we infected about 10 pigs in a farm that was chosen uniformly at random from all the farms. As there are five different pig operation types in our network, we would like to evaluate how each type affect the outbreaks. We run independent sets of simulations where we target a specific operation type (boar stud, farrow, nursery, grower, and market) in each set. We select an operation of that particular type and use it to seed the infection. It is important to note that, the number of pig operations in each type/category is different. The pig population also vary among operations. In our generated network, we have approximately 3.82%, 28.72%, 11.73%, 44.86%, and 10.87% pigs in Boar Stud, Farrow, Nursery, Grower, and Market operations respectively. The outcomes are shown in Fig 5. Here, we define the term 'Epidemic Attack Rate' as,

$$Epidemic~Attack~Rate = \frac{Number~of~pigs~infected~during~the~outbreak}{Total~number~of~pigs} \qquad (1)$$

We find that, the markets are most capable among the five types in spreading the infection while grower and farrow farm types are the second and third most important to consider.

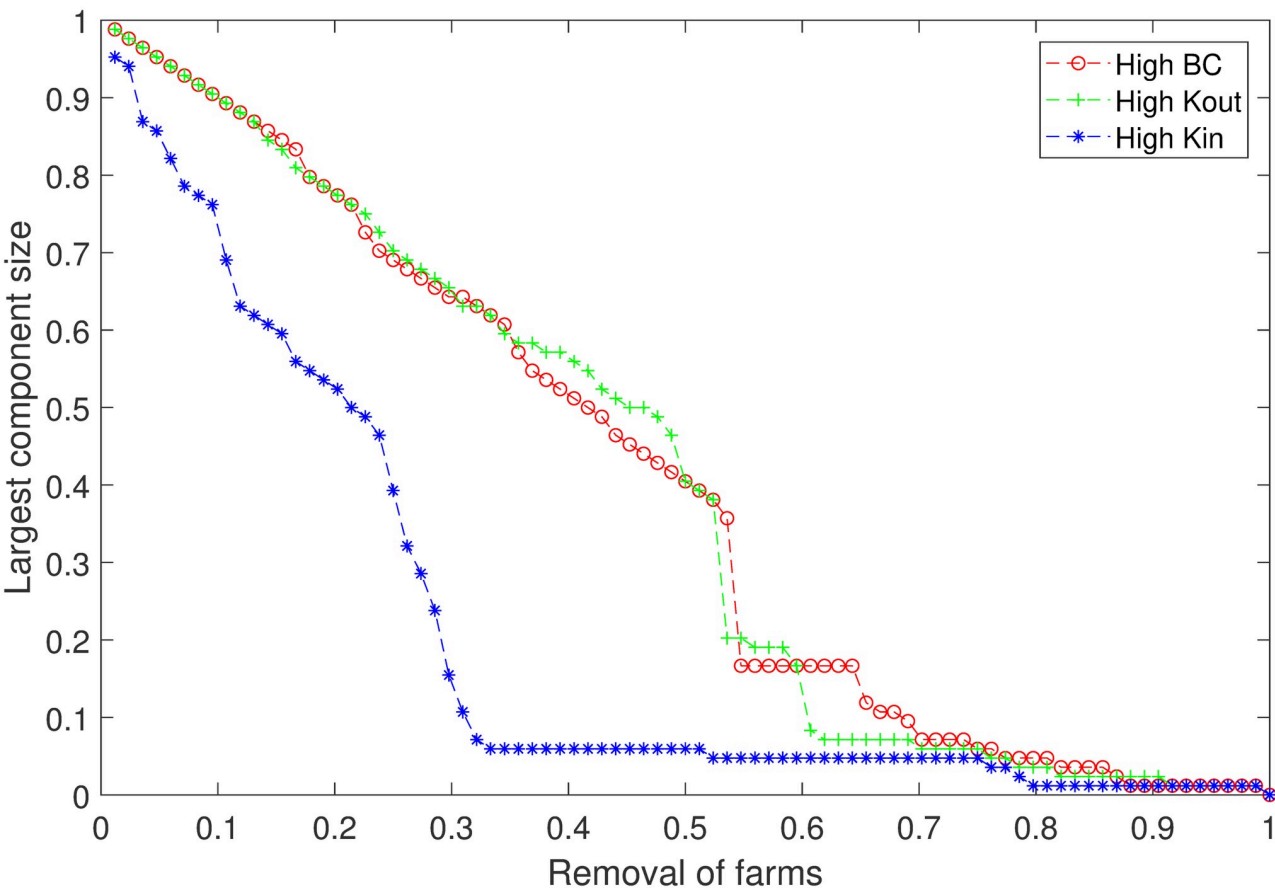

**Fig 3. Network robustness analysis by the gradual removal/isolation of farm nodes.** The farm nodes are removed in a decreasing order of different centrality measures and the size of the largest weakly connected component (at the farm-level) is plotted. Both of the axes are plotted as fractions of total farms in the network. For the removal of nodes, they are separately ranked with three independent centrality measures: high betweenness centrality ($BC$), high out-degree centrality ($K_{out}$), and high in-degree centrality ($K_{in}$).

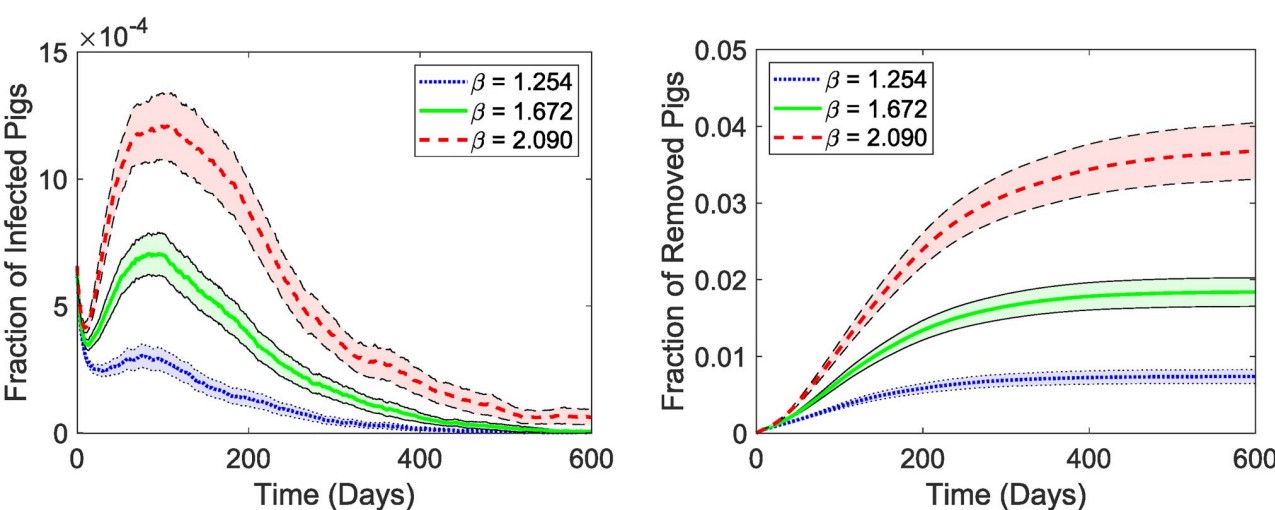

**Fig 4. Time series outbreak results.** Simulated outbreak dynamics in the generated swine network. The results shown above are the averages of 1000 independent simulations. To start each outbreak, a herd/farm was selected uniformly at random where we infected up to 10 pigs which were selected randomly from that particular herd. The simulations were run for three different $\beta$ values (1.672, 1.254, and 2.090) which are shown using different line styles and colors as indicated by the legends. The shaded regions in the plots show 95% confidence intervals. The left plot shows the fraction of infected pigs and the right plot shows the fraction of removed (dead) pigs over time for the generated pig network.

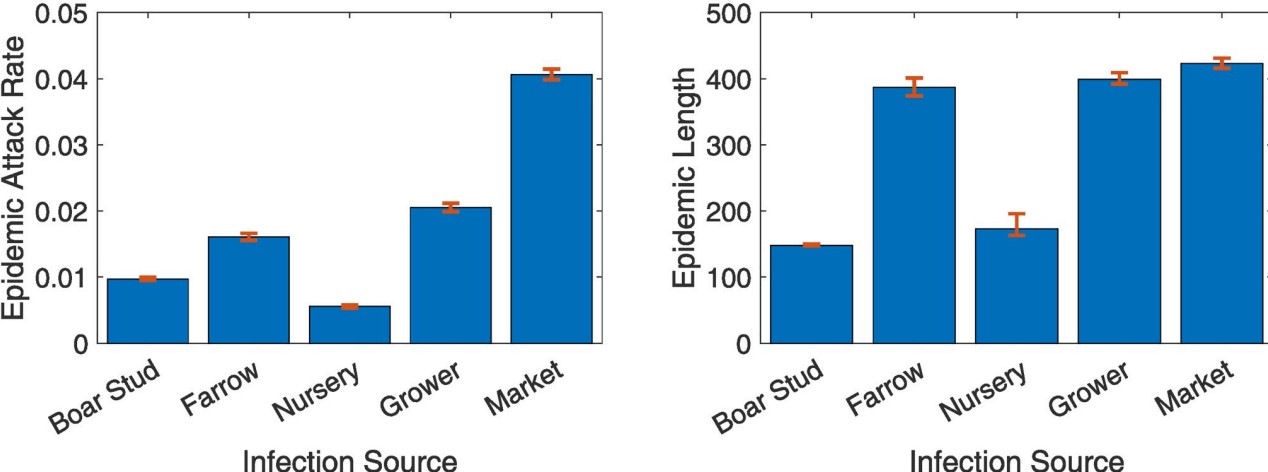

**Fig 5. Outbreak analysis based on source of infection.** Simulated outbreak statistics in the generated swine network. The results shown above are the averages of 10,000 independent simulations. The 95% confidence intervals are shown in red error bars. To start each outbreak, a pig operation was chosen from a given type (either Boar Stud, Farrow, Nursery, Grower, or Market) and up to 10 pigs from that operation were infected. The left plot shows the epidemic attack rates as defined in Eq 1 and the right plot shows the duration of outbreaks.

Although grower farms have 4.13 times the population of the market sites, the market sites cause 1.98 times bigger outbreaks (0.0406 [95% CI 0.0398 0.0414]) compared to grower sites (0.0205 [95% CI 0.0199 0.0212]). Despite that, the duration of the outbreaks caused by the farrow, grower, and market sites are quite comparable (387 [95% CI 374 401], 399 [95% CI 392 409], and 423 [95% CI 416 431] days respectively). The large populations in the grower and farrow farms explain have contributions towards their large outbreaks. Market sites, on the other hand, are potent infection spreaders due to their high connectivity (high in-degree centrality) with remaining farm types.

## Control measures

Due to the lack of cure for African swine fever virus, movement restriction remains a key control method for the policy makers. For this experiment, we use three different network centrality measures (in-degree centrality, $K_{in}$, out-degree centrality, $K_{out}$, and betweenness centrality, $BC$) for the farm nodes and sort the farms in a descending order based on these measures. Next, we gradually place movement restrictions on an increasing number of farms selected from the sorted lists and run outbreak simulations. The attack rates and the outbreak lengths are compared in Fig 6 for three different network centrality measures. Placing movement restrictions based on in-degrees ($K_{in}$) demonstrate the best performance in disease control while restrictions based on betweenness centralities ($BC$) perform the worst. Isolation of top 5 farms based on $K_{in}$ shows about 63.04% [95% CI 61.96 64.13] reduction in the outbreak size (attack rate) and 51.59% [95% CI 50.26 52.91] reduction in outbreak duration compared to the situation without any control measure (Fig 4). For the $K_{out}$ and $BC$ based isolation schemes, we observe 19.6% [95% CI 16.85 21.74] and 4.9% [95% CI 1.63 7.61] reductions respectively in outbreak sizes with 8.5% [95% CI 6.61 11.64] and 6.4% [95% CI 4.5 8.47] reductions respectively in outbreak durations when we isolate 5 farms.

As there is no effective vaccine for ASF, we model hypothetical vaccines with 80% efficacy. This efficacy value has been mentioned in other cases as a nominal requirement to make a vaccine marketable [30]. For our model, it means that, 80 out of 100 vaccinated pigs will be fully

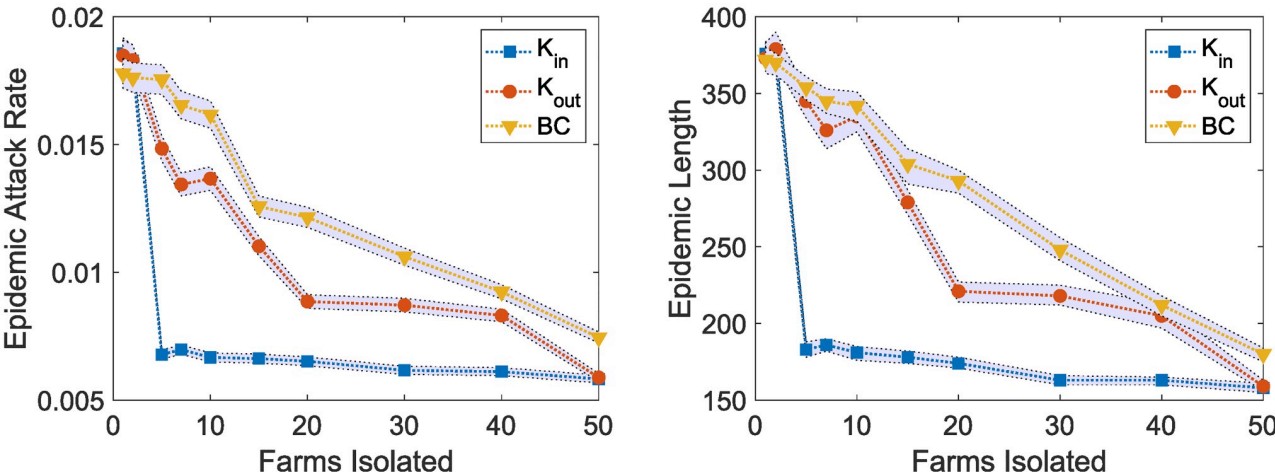

**Fig 6. Comparison of different targeted isolation schemes based on farm node centrality measures.** Three different movement restriction strategies (high in degree, $K_{in}$, high out degree, $K_{out}$, and high betweenness, $BC$) are compared. For each strategy, different number of farms are isolated from a centrality based sorted descending list. The left plot shows the epidemic attack rates and the right plot shows the epidemic lengths. The data points are mean values computed from 10,000 stochastic simulations and the shaded regions show 95% confidence intervals.

immune to the invading pathogen. We use the same set of centrality based sorting strategies to select farms for vaccinations (in-degree $K_{in}$, out-degree $K_{out}$, and betweenness centrality, $BC$ measures). The results are shown in Fig 7. Once again, immunizing farms based on high in-degree ($K_{in}$) is found to be the most effective strategy while immunization based on high betweenness centrality ($BC$) is found to be least effective in disease control. Vaccination of top 5 farms based on $K_{in}$ shows about 59.78% [95% CI 58.70 60.87] reduction in the outbreak size (attack rate) and 44.18% [95% CI 42.86 45.77] reduction in outbreak duration compared to the situation without any control measure (Fig 4). For the $K_{out}$ and $BC$ based immunization

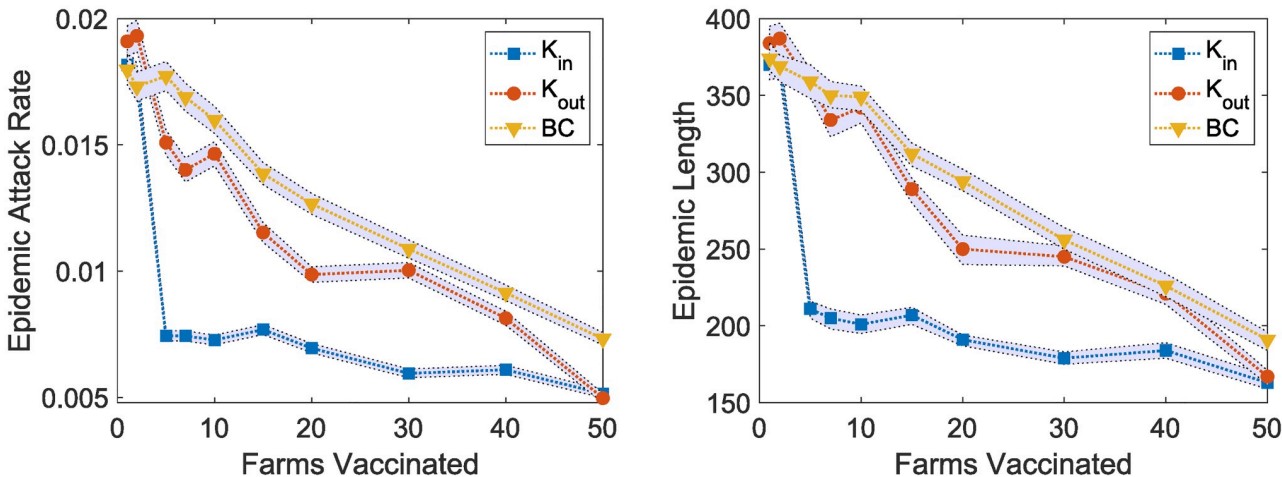

**Fig 7. Comparison of different targeted vaccination schemes based on farm node centrality measures.** Three different vaccination strategies (high in degree, $K_{in}$, high out degree, $K_{out}$, and high betweenness, $BC$) are compared. For each strategy, different number of farms are immunized from a centrality based sorted descending list. The hypothetical vaccines are 80% effective. The left plot shows the epidemic attack rates and the right plot shows the epidemic lengths. The data points are mean values computed from 10,000 stochastic simulations and the shaded regions show 95% confidence intervals.

schemes, we observe 17.93% [95% CI 15.22 20.65] and 3.8% [95% CI 0.54 7.07] reductions respectively in outbreak size with 5.56% [95% CI 2.91 7.67] and 5.03% [95% CI 2.12 7.94] reductions respectively in outbreak duration when we vaccinate 5 farms. The comparative results of the vaccination strategies resemble the results found in the previous experiment for movement restriction measures.

## Conclusion

In this study, we have proposed a method to generate movement networks from available data on the US swine industry, where we have utilized movement network characteristics available for two counties in Minnesota. Using the generated farm-level movement network, we have analyzed multiple centrality properties and performed a robustness analysis to obtain a better insight into the network structure. Using the generated pig-level contact network, we formulated a stochastic *SEIR* model for the transmission of African swine fever. We ran outbreak simulations and examined time-series data with different pig operation types as sources of infection and compared the outcomes. Finally, we analyzed and compared the outcomes of centrality-based targeted isolation and vaccination methods.

The outbreak simulations show that if ASFV is introduced in a random herd, and it is allowed to spread unchecked, it may affect approximately 1.84% of the total swine population with high probability for the two counties in our consideration. Among the five different farm types, infecting the pig population in the market operations causes the most significant outbreaks. The high connectivity of the markets with other farm types and the both-way transmission caused by fomites (e.g., transport vehicles) are the reasons behind such high impact of the markets. The large populations in grower and farrow farm types also make them significant in spreading ASFV infections. Control measures can target these farm types in the event of such outbreaks. In our preliminary farm network analysis, we find that the nursery operations have high out-degrees while the market operations have high in-degrees. We also find that grower operations have high betweenness centrality values. A network robustness analysis reveals that isolating high in-degree farms disrupt the connectivity in the network the most compared to using other centrality measures.

When we examine the impact of centrality-based targeted control measures, the outcomes reinforce our results from the preliminary analysis. We have examined two different control measures with outbreak simulations: movement restriction and hypothetical vaccine. In both cases, we find that controlling farms with high in-degree proves to be beneficial in containing the disease spread. Implementing control in high out-degree farms proves to be slightly better than doing so in high betweenness farms, while both are inferior compared to high in-degree based targeted control. In a separate independent analysis (Fig 5), market operations have proven to be the most potent sources of infections in causing relatively more significant outbreaks compared to other farm types. As the market operations have very high in-degree, our results consistently suggest that these sites should be prioritized in the case of ASFV outbreaks.

Limited public data availability on swine movement in the US compels us to rely on probabilistic network-generation methods to close analytical gaps. Available data on Stevens and Rice counties of Minnesota aided the construction of the movement network. However, these data may be inadequate for the extrapolation of more extensive swine-movement networks. Despite that, our generated network has degree distributions that agree with the given data and the real-world characteristics of the swine production industry. If additional data for movement networks in other locations become available, our network generation algorithms can be used with little or no modifications, depending on the data. We also made a simplifying assumption of having one operation type at a single site, while in practice, there can be

multiple operation types. In addition to that, individual-based simulation models are limited due to computational complexities caused by a large population. Metapopulation models can be a viable solution when considering state-level networks. The network generation techniques can be improved further if more data on swine production operations is made available. Distributed databases could be used to improve traceability and data sharing for the agriculture production supply chain. Further efforts could be made in performing surveys, raising awareness, and motivating the livestock industry to participate in data exchange to support research solutions that can benefit the industry operations.

# Materials and methods

## US swine data

We generate the swine movement network utilizing some of the network characteristics (mixing matrix, in-degree, and out-degree centralities) reported in the Valdes-Donoso [2] paper. The mixing matrix is given in Table 1 and the centralities are shown in Table 2. We define several pig operation types that include farms and markets. Using the operation type distribution described in the same work [2], we classify 5 different pig operations (Boar Stud, Farrow, Nursery, Grower, and Market) as shown in Table 3. The operation types are defined below,

- **Boar Stud**. These farms are used to keep male boars for breeding.

- **Farrow**. Sows are moved to these farrowing farms to give birth (farrow). Piglets stay here up to 3 weeks.

- **Nursery**. Piglets are moved to nursery after weaning where they could stay up to 8 weeks.

- **Grower**. Pigs are moved from nursery to grower/finisher farms where they will gain market weight at about six months of age.

- **Market**. The market type includes buying stations and/or slaughter plants.

**Table 1. Mixing matrix (probability of movement from row type to column type) for swine movement network** [2]. The pig operation types are abbreviated as B (Boar Stud), F (Farrow), N (Nursery), G (Grower), and M (Market).

|   | B | F | N | G | M |
|---|---|---|---|---|---|
| B | 0.00 | 0.00 | 0.00 | 0.00 | 0.01 |
| F | 0.00 | 0.03 | 0.04 | 0.09 | 0.10 |
| N | 0.00 | 0.00 | 0.00 | 0.13 | 0.00 |
| G | 0.01 | 0.10 | 0.00 | 0.07 | 0.40 |
| M | 0.00 | 0.00 | 0.00 | 0.00 | 0.02 |

**Table 2. Movement network degree centrality data** [2].

|   |   | B | F | N | G | M |
|---|---|---|---|---|---|---|
| In-degree | Average | 0.67 | 0.92 | 0.77 | 1.05 | 11.73 |
|  | SE | 0.67 | 0.14 | 0.1 | 0.07 | 3.59 |
|  | Max | 2 | 5 | 2 | 5 | 57 |
| Out-degree | Average | 1.00 | 2.08 | 3.07 | 1.74 | 0.46 |
|  | SE | 0 | 0.26 | 0.62 | 0.15 | 0.18 |
|  | Max | 1 | 8 | 12 | 12 | 3 |

**Table 3. Pig operation type distribution.**

| Boar Stud(B) | Farrow(F) | Nursery(N) | Grower(G) | Market(M) |
|---|---|---|---|---|
| 1.27% | 27% | 12.66% | 51.9% | 7.17% |

**Table 4. Distribution of pigs in Stevens and Rice counties of Minnesota.**

| Farm Size | No. of Farms | No. of Pigs |
|---|---|---|
| 1 to 24 | 17 | 204 |
| 25 to 49 | 0 | 0 |
| 50 to 99 | 0 | 0 |
| 100 to 199 | 2 | 300 |
| 200 to 499 | 3 | 700 |
| 500 to 999 | 11 | 7,904 |
| 1,000+ | 51 | 240,042 |
| Total | 84 | 249,150 |

Obtaining data from United States Department of Agriculture—National Agricultural Statistics Service (USDA-NASS) [31], we find that two counties (Rice & Stevens) of Minnesota have 84 farms and 249,150 pigs in total. We take the 84 farms and as the operation types are unknown, assign types randomly based on the distribution shown in Table 3.

Availability of operation type distribution data is incomplete as well, there are several suppressed data fields. We allot pigs in those unknown fields randomly and make sure that the aggregate statistics are maintained. The adjusted combined statistics for Stevens and the Rice counties are provided in Table 4.

While the USDA-NASS data provide the total number of farms and pigs in a size class, it is impossible to infer the number of pigs at individual farms. Hence, we use a random allocation mechanism to assign the number of pigs for each farm while maintaining the aggregate statistics of Table 4. Once we generate the network edges, we assign a weight to them to indicate amount/rate of movement via that edge. According to the work of Spencer R. Wayne [32], the Rice and the Stevens counties experience mean shipment of 21 and 15 per year and median shipment of 10 and 7 per year respectively. Based on those values, our combined network is estimated to have mean shipment of 17.38 per year and median shipment of 8.5 per year. We use lognormal distribution and assign randomly generated shipment rate values to network links.

### Network terminology

We use several network structure and analysis related terminologies throughout this paper. These terminologies are described below,

- **Network/Graph**. A network (also called graph) is a structure consisting of nodes (also called vertices) and links (also called edges). A link connects two vertices and it can be either directed or undirected.

- **Stub**. A stub is half a link. It's a link with a node on one end and an empty handle on the other end. Empty handles of two stubs can be joined together to form the link and thus create a connection between two nodes.

- **Path, Shortest Path**. A path is a sequence of links which joins a sequence of vertices which are all distinct. A shortest path is the minimum length path between two nodes in a network.

- **Connected Component**. A connected component (also referred to as a component) is a subset of nodes where there is a path between every pair of nodes in that subset. Two distinct components aren't connected by any path. If all nodes in a component are connected via bidirectional paths then the component is strongly connected, otherwise it is called weakly connected (path in one direction). In this paper, we consider weakly connected components as transmission can happen in the reverse direction of the animal movement via fomites (e.g. transport vehicles).

We use several centrality measures to determine the importance of the nodes. The centrality measure can quantitatively characterize how important a node is in the network.

- **Degree Centrality**. The degree ($K$) of a node is the number of links associated with that node. In case of directed networks, we define in-degree ($K_{in}$) as the number of links going into the node and out-degree ($K_{out}$) as the number of links coming out of the node.

- **Betweenness Centrality**. There is a shortest path for every pair of nodes in a connected component. The betweenness centrality ($BC$) of a node is the total number of shortest paths that pass through that node (not counting the paths starting from or ending at that node).

## Network generation

The swine network is synthesized using the available swine farm and movement related data described in the previous section. The network generation process is completed in several stages:

1. Assign each farm node a single operation type randomly based on the farm type distribution given in Table 3.

2. Assign directed in and out-degree values or handles (stubs) to each farm node randomly based on the degree distribution given in Table 2.

3. Connect out-handle (stub) of a farm node to in-handle (stub) of another farm node randomly, based on the mixing matrix given in Table 1.

4. Assign shipment rate values to all the directed links from a lognormal distribution with the obtained mean and the median shipment rate values.

5. Assign each farm a certain number of pigs randomly, based on the distribution given in Table 4.

6. Generate the within-farm undirected contact links among the pigs based on the Erdös—Rényi process with 50% probability.

7. Convert the shipment rates of farm links into probabilities and generate between-farm undirected contact links for the pigs based on those rates.

We generate a farm level movement network at step 4 and a pig level contact network at step 7. It is necessary to mention that, working with a graph that has 249,149 nodes, is computationally intractable due to the large number of within-farm links among the pigs. Hence, we scale down the pig population by a constant factor of 20, which makes the network small enough to be computationally feasible, while retaining sufficient pig nodes to maintain connectivity properties of the farm level network. As a consequence, most of our ASF model results are qualitative investigations of outbreak behavior.

## ASFV epidemic model

Our network based epidemic model is shown in Fig 8. Using the farm level movement network, we generate a pig level movement network. In this network, each node is an individual pig and the links connecting a node to other nodes indicate interactions with other pigs (nodes). A pig has a lot more links to other pigs within the same farm compared to pigs which are at other farms. The links to other farms are generated based on the movement network. In Fig 8, a host node (pig) is marked using a solid circle and the links to other nodes are marked by the solid lines. A host (pig) can get exposed from any of its infected neighbors at the rate of $\beta$, which is defined as the infection rate. For modeling African swine fever infection dynamics, we divide the pig population into four groups: Susceptible ($S$), Exposed ($E$), Infected ($I$), and Removed/Dead ($R$). The healthy pigs which are free from ASF infection are classified as Susceptibles. If such a healthy pig comes into contact with infected pigs containing the virus, it may get infected at the rate $\beta Y_i(t)$, where $Y_i(t)$ is the number of infected neighbors of node $i$ at time $t$. If the transmission of pathogen occurs, a healthy pig enters into the Exposed group where it stays for the duration of the incubation period. On average, this period is denoted by $1/\sigma$. Once it shows symptoms, it moves into the Infected group. It stays there for an average time of $1/\gamma$ before it is removed. As for ASF, the mortality is assumed to be 100% and no pig recovers. Hence, all infected pigs die at the end of the infected period. However, in multiple cases for our simulations, we will hypothetically vaccinate pigs. Based on the vaccine efficacy, alive pigs may move to the removed class too.

The model parameters are shown in Table 5. The last column in this table mentions the different sources from where we obtained the parameter values. For $\beta$, we used estimated data from [28] where median transmission rate values were computed for 9 herds. These values are listed in Table 6. We take the weighted median from this set of data and use that β value in our simulations. We use the well-developed GEMFsim [33] tool to run our simulations.

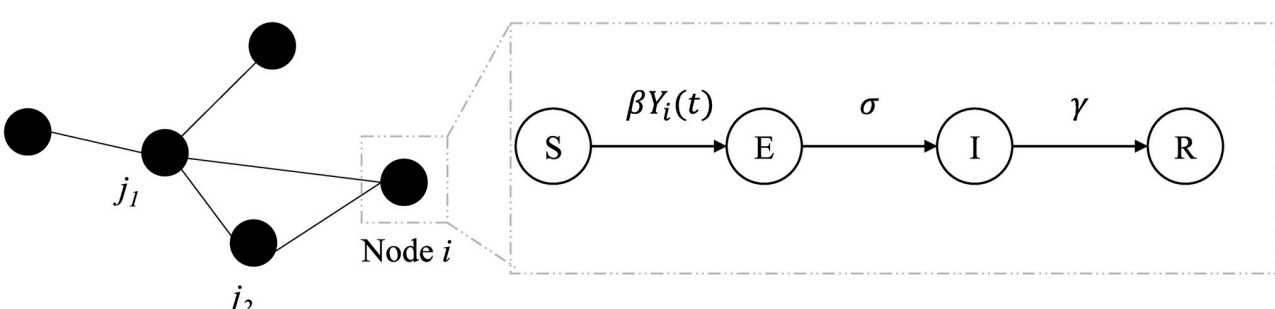

**Fig 8. ASF epidemic model.** The network based *SEIR* epidemic model for African swine fever virus. The black solid circles indicate host nodes (individual pigs) and the solid lines connecting them indicate contacts (direct or fomites) that can act as infection pathways of ASFV. Each node can be in any of the four states, Susceptible ($S$), Exposed ($E$), Infected ($I$), or Recovered ($R$). The rates at which a host can move from one state to another are indicated by the parameters (See Table 5) adjacent to corresponding arrows. Here, $Y_i(t)$ is the number of infected contacts of node $i$ at time $t$.

**Table 5. ASFV epidemic model parameters.**

| Symbol | Definition | Range | Value | Reference |
|---|---|---|---|---|
| $\beta$ | Transmission Rate | 0.7–2.2 | 1.6719 | [28] [29] |
| $1/\sigma$ | Latent Period | - | 7.78 | [28] |
| $1/\gamma$ | Infectious Period | - | 8.3 | [28] |

**Table 6. Transmission rate estimated for 9 pig herds by Guinat et al. [28].**

| Herd Size | 1614 | 1949 | 1753 | 1833 | 1320 | 600 | 600 | 600 | 2145 |
|-----------|------|------|------|------|------|-----|-----|-----|------|
| $\beta$ | 2 | 1 | 2.2 | 0.7 | 1.6 | 2.1 | 1.6 | 2.2 | 0.8 |

## Supporting information

**S1 File. FarmNodeList.** List of farm nodes with their operation types.
(TXT)

**S2 File. FarmEdgeList.** List of farm links and their weights.
(TXT)

**S3 File. PigsNodeList.** List of pig nodes and the farms they belong to.
(TXT)

**S4 File. PigsEdgeList.** List of pig links and their weights (all are equally weighted to 1).
(TXT)

**S1 Appendix. Appendix A: Network generation algorithms.**
(PDF)

## Acknowledgments

We would like to thank Dr. Michael W. Sanderson for his useful insights into the swine farming industry. We would also like to thank the developers of the GEMFsim [33] tool which was used to run outbreak simulations.

## Author Contributions

**Conceptualization:** Tanvir Ferdousi, Sifat Afroj Moon, Caterina Scoglio.

**Data curation:** Tanvir Ferdousi, Sifat Afroj Moon.

**Formal analysis:** Tanvir Ferdousi, Sifat Afroj Moon, Adrian Self, Caterina Scoglio.

**Funding acquisition:** Adrian Self, Caterina Scoglio.

**Investigation:** Tanvir Ferdousi, Adrian Self, Caterina Scoglio.

**Methodology:** Tanvir Ferdousi, Sifat Afroj Moon.

**Project administration:** Adrian Self, Caterina Scoglio.

**Resources:** Caterina Scoglio.

**Software:** Tanvir Ferdousi.

**Supervision:** Adrian Self, Caterina Scoglio.

**Validation:** Tanvir Ferdousi, Sifat Afroj Moon, Adrian Self, Caterina Scoglio.

**Visualization:** Tanvir Ferdousi, Sifat Afroj Moon.

**Writing – original draft:** Tanvir Ferdousi, Sifat Afroj Moon.

**Writing – review & editing:** Tanvir Ferdousi, Sifat Afroj Moon, Adrian Self, Caterina Scoglio.

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
