## [Decision Letter · Decision Letter 0]

9 Aug 2019

PONE-D-19-17842

Generation of swine movement network and analysis of efficient mitigation strategies for African swine fever virus

PLOS ONE

Dear Ferdousi,

Thank you for submitting your manuscript to PLOS ONE. After careful consideration, we feel that it has merit but does not fully meet PLOS ONE’s publication criteria as it currently stands. Therefore, we invite you to submit a revised version of the manuscript that addresses the points raised during the review process.

The paper should be reformatted, clearly separating the results from the methods. The stochasticity and the variation in the results should be clearly incorporated in the study, and analyzed with appropriate statistical tools.  The other observations of the reviewers are also valid and should be addressed.

We would appreciate receiving your revised manuscript by Sep 23 2019 11:59PM. To enhance the reproducibility of your results, we recommend that if applicable you deposit your laboratory protocols in protocols.io, where a protocol can be assigned its own identifier (DOI) such that it can be cited independently in the future. For instructions see: http://journals.plos.org/plosone/s/submission-guidelines#loc-laboratory-protocols

We look forward to receiving your revised manuscript.

Kind regards,

Willem F. de Boer

Academic Editor

PLOS ONE

Reviewers' comments:

Reviewer's Responses to Questions

**Comments to the Author**

1. Is the manuscript technically sound, and do the data support the conclusions?

Reviewer #1: Partly

Reviewer #2: Yes

2. Has the statistical analysis been performed appropriately and rigorously? 

Reviewer #1: No

Reviewer #2: Yes

3. Have the authors made all data underlying the findings in their manuscript fully available?

Reviewer #1: Yes

Reviewer #2: Yes

4. Is the manuscript presented in an intelligible fashion and written in standard English?

Reviewer #1: Yes

Reviewer #2: Yes

5. Review Comments to the Author

Reviewer #1: The manuscript presents a model to reconstruct the contact network between swine production herds in the Rice and Stevens counties (84 herds) given summary statistics available and previously published. This network was designed at two scales corresponding to between-herd shipments and between-animal contacts. An epidemic model was then superposed on the generated network to analyze the spread of an infectious disease (namely ASF) on the net. The outcomes show a dramatic transmission process reaching 80% of the pig population. Finally, the authors chased to represent control measure based on vaccination.

As mentioned in the introduction, the analysis of infectious diseases on networks is very important and could help designing targeted control measures. In that view the paper is of interest. However, I have some concerns concerning the form and contain of the manuscript.

1. The structure of the manuscript was a first surprise, but there might be a bias due to the different expertise field from the authors and the reviewer. Being more oriented on the epidemiological field, I am effectively more used to classical papers with material and methods clearly separated form results. Here each development part is directly followed by the related results making the different subsection relatively independent. As such, the paper rather appears to me as a working document than a scientific paper.

2. The choice of representation of each individual pig is questionable. Indeed, the graph with 249149 nodes was intractable (L 132) and the number of pigs was therefore rescaled by a factor 20 (a little bit more than 10000 pigs were considered). The connectivity of the farms are driven by the in and out-degrees defined in table 1, so I understand that the structure of network was not impacted, however with a Erdos-Renyi model, the pig-network centrality measures are undoubtedly modified and the consequences on infection dynamics could be dramatically modified.

3. Network analysis: The authors retrieve their inputs. Although it is worth verifying before going further, this is not a result and again appears more like a verification for a work report.

4. The figures 1-4 are relatively redundant. Maybe boxplots representing the distributions of the centrality measures would be more appropriate.

5. I do not really understand the role of the section “network robustness analysis”. The authors want to see the impact disruption of network connectivity on disease spread. Why not doing so with the epidemiological model implemented? The results would be from far more interesting, especially for ASF since it is the only mtype of measures actually feasible at that time.

6. ASF epidemic model, a SEIR model was used accounting for contacts between individual pigs. The model is implemented using GEMFsim,meaning that the model is stochastic,isn’t it? I’ll come back to that point in the next comments. The parameter estimates show very wide variation interval, especially for the transmission rate in the different studies. Haw was chosen the values in the model? what if other choice had been made?

7. Temporal dynamics. First, concerning the initial condition. The authors introduce ASF into their network consitdering 5% of infected pigs randomly selected among the herds (representing 500 animals due to the scaling factor and up the 12000 in real life!). This means that the initial number of herds that were initially infected is not fixed and could be rather high. With a transmission rate of 1.6719 (impressive precision), and a Erdos-Renyi network within farms, it is highly likely the whole population in a herd get rapidly infected, partially explaining the 82% of infected animals. Then the authors considered introduction in specific farm types, but keeping the 5% of initially infected pigs. I am still wondering how many herds were initially infected. Whatever the assumption, it is (hopefully) unlikely that many herds seed the infection at the same time and the transmission process from farm to farm is the major risk. Maybe selection of one farm as a seeder and monitoring within-herd and between-herd spread would have been better.

8. The achievement of 80% cumulative incidence is not reliable for a disease like ASF. Indeed as mentioned in the introduction for the Chinese situation (and the same occurs in Europe), the detection of ASF leads to movement restriction and herd stamping out. This is why testing the robustness of the network should be considered as a control measure and not a validation of the model. Several studies already considered such measures in Europe…

9. The authors define the attack rate and I come back here to the stochasticity. All results are given as averages and only compared as such. The authors should consider to highlight the variability in their results with adapted statistical tests to make clear comparisons of their outcomes.

10. Instead of testing realistic control measures for ASF, the authors decided to test the effect of an hypothetical vaccine with 80% efficacy. Again the variability and significance of the obtained difference is lacking. This does not allow for clear conclusions.

11. Lines 130 steps c and g should be replaced by 3 and 7.

In conclusion, despite the media and scientific interest on ASF, I think the model is not well designed for such application. The assumptions are somewhere unlikely and the conclusions not relevant for this disease. Maybe an application to swine influenza would have been worth. I also underlined the lack of statistical tests to analyze the variability of model outcome to derive conclusions. The network generation is interesting in itself, but the representation of individual pigs and their relative interactions need to be clearly justified. I recommend to re-submit a new draft revised in depth to obtain an epidemiological model suitable for gaining insight on infectious diseases transmission.

Reviewer #2: Comments to the Author

This is a well-written, and important study providing insights for using animal movement networks to imply targeted disease control practices. This study generated a pig farm network in US to simulate the spread of African swine fever virus and to identify critical spreaders in the network, which is insightful for planning targeted surveillance and vaccination.

I believe this manuscript has a potential for publication. The manuscript is well organized and the methods are sound. However, more explanations here and there are needed to facilitate better understanding and make the methodology repeatable. Below are some issues of a fairly minor nature that need revision.

L23 – ‘stub’ is not a clear word for me, need a definition or replace with ‘weight’?

L23 – In early August, but which year?

L52 – From here until the end of this paragraph, previous studies about ASFV were listed but not discussed. It would be better to summarize the information and show the knowledge gaps. One sentence about the relationships about the reviewed studies and this study is worthwhile.

L92 – It is confusing to use ‘sites’ here. Does it mean ‘farm types’ or ‘operation types’? Better give a definition when you mention it for the first time and keep them consistent throughout the manuscript.

L95 – Why these two counties?

L105 – It is not clear how the edges were generated. The previous part of this paragraph only mentioned how the nodes were defined. The details about defining an edge in the movement network should be added.

L116 – It would be better to add some explanations of these five operation types to help understand the ecological backgrounds.

L118 – Again, ‘handles (stubs)’ is not clear.

L122 – With lognormal distribution?

L131 – ‘step c’ and ‘step g’ were not defined.

L142, 146 – The in-degree and out-degree values were assigned randomly according to stage 2 in ‘Network Generation’, so the results of in-degree and out-degree centralities were totally dependent on how you assigned them. I suggest to either delete these results about degree centrality or add discussions about the risk of circular arguments.

L165 – How the ‘largest connected component’ was selected? Is this metric representative in measuring the connectivity or risk of disease transmission in a network?

L284 – It is worth discussing the applicability of this method for the other counties. If it is applicable for other regions, the simplified survey methods can save the expenses for disease surveillance.

Comments on tables:

Table 1 – Abbreviations should be explained in the table legends.

Figure 9 – How the percentages of vaccination were defined? A sensitivity analysis would be more profound than using these random numbers.

6. PLOS authors have the option to publish the peer review history of their article (what does this mean?). If published, this will include your full peer review and any attached files.

Reviewer #1: No

Reviewer #2: Yes: Yanjie Xu

---

## [Author Response · Author response to Decision Letter 0]

16 Oct 2019

Reviewer #1: The manuscript presents a model to reconstruct the contact network between swine production herds in the Rice and Stevens counties (84 herds) given summary statistics available and previously published. This network was designed at two scales corresponding to between-herd shipments and between-animal contacts. An epidemic model was then superposed on the generated network to analyze the spread of an infectious disease (namely ASF) on the net. The outcomes show a dramatic transmission process reaching 80% of the pig population. Finally, the authors chased to represent control measure based on vaccination.

As mentioned in the introduction, the analysis of infectious diseases on networks is very important and could help designing targeted control measures. In that view the paper is of interest. However, I have some concerns concerning the form and contain of the manuscript.

1. The structure of the manuscript was a first surprise, but there might be a bias due to the different expertise field from the authors and the reviewer. Being more oriented on the epidemiological field, I am effectively more used to classical papers with material and methods clearly separated form results. Here each development part is directly followed by the related results making the different subsection relatively independent. As such, the paper rather appears to me as a working document than a scientific paper.

Response: Thank you for your valuable insight. We have restructured the contents of the paper so that the manuscript adheres to the standard formats followed for scientific papers. We have completely isolated the results from the Materials and Methods section and compiled those in the Results section. The generated network is one of the results of this work, hence it is now moved to the Results section along with the preliminary analyses.

2. The choice of representation of each individual pig is questionable. Indeed, the graph with 249149 nodes was intractable (L 132) and the number of pigs was therefore rescaled by a factor 20 (a little bit more than 10000 pigs were considered). The connectivity of the farms are driven by the in and out-degrees defined in table 1, so I understand that the structure of network was not impacted, however with a Erdos-Renyi model, the pig-network centrality measures are undoubtedly modified and the consequences on infection dynamics could be dramatically modified.

Response: Thank you for bringing this point. Due to computational limitations it is not practical for us to work with such large number of nodes. We resorted to this scaling which do not affect the farm movement network structure, which is a key aspect of our model. The within farm interactions between pigs are modeled as Erdos-Renyi with a certain probability of having a connection between any pair of nodes (pigs). This connection probability remains the same despite the scaling. As the degree distribution in Erdos-Renyi is binomial (approximately Poisson for large number of nodes) and we keep the probability (p) same while reducing the number of nodes (n), the shape of the distribution doesn’t change. We attempt to capture the essential properties in our scaled network and present the result as fractions of total population instead of actual numbers (for example, Fig 4).

3. Network analysis: The authors retrieve their inputs. Although it is worth verifying before going further, this is not a result and again appears more like a verification for a work report.

Response: Thank you for your feedback. During the restructuring of the manuscript we discarded most of this section. We have developed the network generator as a major contribution of this work and due to this reason, consider the generated network to be a result. Hence, we included this network in the Results section along with some preliminary analyses of centrality and robustness. The outcomes of these analyses are relevant in explaining the results in Outbreak Dynamics and Control Measures subsections.

4. The figures 1-4 are relatively redundant. Maybe boxplots representing the distributions of the centrality measures would be more appropriate. 

Response: Thank you for the feedback. We have removed Fig 2-4 from the previous version and kept only Fig 1 in the new version. The centrality analysis is included as boxplots (Fig 2 in the new version).

5. I do not really understand the role of the section “network robustness analysis”. The authors want to see the impact disruption of network connectivity on disease spread. Why not doing so with the epidemiological model implemented? The results would be from far more interesting, especially for ASF since it is the only mtype of measures actually feasible at that time.

Response: Thank you for the discussion. The robustness analysis tells us what we can infer from the network structure before even going into a disease simulation. In the new version, we have added a movement restriction based control measure with the epidemiological model implemented. This can be found in the Control Measures subsection (Fig 6).

6. ASF epidemic model, a SEIR model was used accounting for contacts between individual pigs. The model is implemented using GEMFsim,meaning that the model is stochastic,isn’t it? I’ll come back to that point in the next comments. The parameter estimates show very wide variation interval, especially for the transmission rate in the different studies. Haw was chosen the values in the model? what if other choice had been made?

Response: Thank you for raising an important question. Based on the works of Guinat et al. (2017) on Inferring within-herd transmission parameters for African swine fever virus using mortality data from outbreaks in the Russian Federation, we took the median value for the range of transmission rate (β) estimated for 9 different herds from that work. With those 9 estimations, we computed a weighted median of the transmission rate which is given in Table 5. In this new version, we have added an explanation in the ASFV Epidemic Model subsection along with a table (Table 6) of the data obtained. We have also tested with transmission rates 25% above and below the median and compared the results in the Outbreak Dynamics subsection (Fig 4).

[Lines 353 - 356]

For β, we used estimated data from [28] where median transmission rate values were computed for 9 herds. These values are listed in Table 6. We take the weighted median from this set of data and use that β value in our simulations.

7. Temporal dynamics. First, concerning the initial condition. The authors introduce ASF into their network consitdering 5% of infected pigs randomly selected among the herds (representing 500 animals due to the scaling factor and up the 12000 in real life!). This means that the initial number of herds that were initially infected is not fixed and could be rather high. With a transmission rate of 1.6719 (impressive precision), and a Erdos-Renyi network within farms, it is highly likely the whole population in a herd get rapidly infected, partially explaining the 82% of infected animals. Then the authors considered introduction in specific farm types, but keeping the 5% of initially infected pigs. I am still wondering how many herds were initially infected. Whatever the assumption, it is (hopefully) unlikely that many herds seed the infection at the same time and the transmission process from farm to farm is the major risk. Maybe selection of one farm as a seeder and monitoring within-herd and between-herd spread would have been better.

Response: Thank you for the suggestion. Previously, we were randomly infecting among herds of the entire network or from specific farm types. We agree with your comment that it is unlikely that multiple herds will start with the infection at the same time. In this current version, we have changed those initial conditions. In our new simulation model, we randomly choose a herd and infect a small number of pigs in that herd. All the results have been updated based on this. Due to this, there is a significant reduction in the fraction of infected animals and we have updated some of our conclusions accordingly.

8. The achievement of 80% cumulative incidence is not reliable for a disease like ASF. Indeed as mentioned in the introduction for the Chinese situation (and the same occurs in Europe), the detection of ASF leads to movement restriction and herd stamping out. This is why testing the robustness of the network should be considered as a control measure and not a validation of the model. Several studies already considered such measures in Europe…

Response: Thank you for your comment. In the current version, we have implemented centrality based targeted movement restriction/ isolation measures in our model and we have added some new results (Fig 6).

9. The authors define the attack rate and I come back here to the stochasticity. All results are given as averages and only compared as such. The authors should consider to highlight the variability in their results with adapted statistical tests to make clear comparisons of their outcomes.

Response: Thank you for your important suggestion. We have computed 95% confidence intervals in all our disease simulation averages and all the figures have been updated to reflect variabilities (Fig 4 - 7). As an example, please refer to Fig 6 that we included in this document in response to your comment/point no. 8. 

10. Instead of testing realistic control measures for ASF, the authors decided to test the effect of an hypothetical vaccine with 80% efficacy. Again the variability and significance of the obtained difference is lacking. This does not allow for clear conclusions.

Response: Thank you for your suggestion. We have added targeted movement restriction measures (Fig 6) and compared with targeted hypothetical vaccination (Fig 7) measures. We have also added 95% confidence intervals in those results (Fig 6, 7 in the current version).

11. Lines 130 steps c and g should be replaced by 3 and 7.

Response: Thank you. We have fixed the error.

In conclusion, despite the media and scientific interest on ASF, I think the model is not well designed for such application. The assumptions are somewhere unlikely and the conclusions not relevant for this disease. Maybe an application to swine influenza would have been worth. I also underlined the lack of statistical tests to analyze the variability of model outcome to derive conclusions. The network generation is interesting in itself, but the representation of individual pigs and their relative interactions need to be clearly justified. I recommend to re-submit a new draft revised in depth to obtain an epidemiological model suitable for gaining insight on infectious diseases transmission.

Reviewer #2: Comments to the Author

This is a well-written, and important study providing insights for using animal movement networks to imply targeted disease control practices. This study generated a pig farm network in US to simulate the spread of African swine fever virus and to identify critical spreaders in the network, which is insightful for planning targeted surveillance and vaccination.

I believe this manuscript has a potential for publication. The manuscript is well organized and the methods are sound. However, more explanations here and there are needed to facilitate better understanding and make the methodology repeatable. Below are some issues of a fairly minor nature that need revision.

L23 – ‘stub’ is not a clear word for me, need a definition or replace with ‘weight’?

Response: Thank you for raising this confusion. A stub in the context of graph theory is an open ended handle, that is, an incomplete link with a node on one end and the other end being empty/open. Once it connects to another stub of a node, a link is created. The term ‘weight’ is not fully appropriate in our case. We have added a definition of stub in our newly added Network Terminology subsection.

[Lines 280 - 282]

Stub. A stub is half a link. It's a link with a node on one end and an empty handle on the other end. Empty handles of two stubs can be joined together to form the link and thus create a connection between two nodes.

L23 – In early August, but which year?

Response: Thank you. We have corrected that. It would be August 2018.

[Lines 45 - 47]

They reported their first outbreak in early August 2018 and since then there have been about 158 outbreaks in 32 provinces [19].

L52 – From here until the end of this paragraph, previous studies about ASFV were listed but not discussed. It would be better to summarize the information and show the knowledge gaps. One sentence about the relationships about the reviewed studies and this study is worthwhile.

Response: Thank you, we have added the following lines to help clarify the knowledge gaps.

[Lines 69 - 72]

Most of the ASFV research is focused on parameter estimates while several others investigate virus importation risk in US mainland. Despite the numerous studies, there is a lack of knowledge on how the swine industry in the US would be affected in case an ASFV outbreak starts in the US.

L92 – It is confusing to use ‘sites’ here. Does it mean ‘farm types’ or ‘operation types’? Better give a definition when you mention it for the first time and keep them consistent throughout the manuscript.

Response: We have replaced the term sites with operations. We have market operations and farm operations (Boar Stud, Farrow, Nursery, and Grower). The term operation is used as general term while an operation can be a market or a farm.

[Lines 243 - 245]

We define several pig operation types that include farms and markets. Using the operation type distribution described in the same work [2], we classify 5 different pig operations (Boar Stud, Farrow, Nursery, Grower, and Market) as shown in Table 3.

L95 – Why these two counties?

Response: Thank you for this question. Due to privacy concerns, detailed data for swine movement in the USA is not available. However, summary data of the movement network for these two counties were publicly available due to the studies performed by the UMN group [2, 32]. This data was used to reconstruct the network for these two counties of Minnesota.

L105 – It is not clear how the edges were generated. The previous part of this paragraph only mentioned how the nodes were defined. The details about defining an edge in the movement network should be added.

Response: Summary of the steps for edge generation are given in the Network Generation subsection steps 2 and 3. The details (pseudocode) of the edge generation process is given in Algorithm 2: F_GRAPH_GEN. We allot stubs to each node based on the degree centrality data (Table 2). Later we join the stubs based on the mixing matrix in (Table 1).

L116 – It would be better to add some explanations of these five operation types to help understand the ecological backgrounds.

Response: Thank you. We have added some explanation of the five operation types in the US Swine Data subsection.

[Lines 247 - 254]

• Boar Stud. These farms are used to keep male boars for breeding.

• Farrow. Sows are moved to these farrowing farms to give birth (farrow). Piglets stay here up to 3 weeks.

• Nursery. Piglets are moved to nursery after weaning where they could stay up to 8 weeks.

• Grower. Pigs are moved from nursery to grower/finisher farms where they will gain market weight at about six months of age.

• Market. The market type includes buying stations and/or slaughter plants.

L118 – Again, ‘handles (stubs)’ is not clear.

Response: As mentioned before, we have added a definition of stub in our newly added Network Terminology subsection.

L122 – With lognormal distribution?

Response: Thank you. We have corrected this line.

[Lines 315 - 316]

Assign shipment rate values to all the directed links from a lognormal distribution with the obtained mean and the median shipment rate values.

L131 – ‘step c’ and ‘step g’ were not defined.

Response: Thank you, we have corrected this.

[Lines 323 - 324]

We generate a farm level movement network at step 4 and a pig level contact network at step 7.

L142, 146 – The in-degree and out-degree values were assigned randomly according to stage 2 in ‘Network Generation’, so the results of in-degree and out-degree centralities were totally dependent on how you assigned them. I suggest to either delete these results about degree centrality or add discussions about the risk of circular arguments.

Response: Thank you for your suggestion. The data was only sampled randomly, but it was based on a distribution given in Table 2. Hence, the computation/validation is redundant here in terms of result. We have removed the separate centrality plots. However, centrality is relevant in our targeted disease control measure experiments (Control Measures subsection). Hence a discussion was kept in the Movement Network subsection.

L165 – How the ‘largest connected component’ was selected? Is this metric representative in measuring the connectivity or risk of disease transmission in a network?

Response: Thank you for the question. Typically, we measure all the connected components of the network and take the size of the largest one. It indicates how fragmented the network is. If the component size is small, it means the network is broken into many small fragments (where disease cannot spread from one to another). To clarify how components are computed we added a definition in the Network Terminology subsection.

[Lines 286 - 293]

Connected Component. A connected component (also referred to as a component) is a subset of nodes where there is a path between every pair of nodes in that subset. Two distinct components aren't connected by any path. If all nodes in a component are connected via bi-directional paths then the component is strongly connected, otherwise it is called weakly connected (path in one direction). In this paper, we consider weakly connected components as transmission can happen in the reverse direction of the animal movement via fomites (e.g. transport vehicles).

L284 – It is worth discussing the applicability of this method for the other counties. If it is applicable for other regions, the simplified survey methods can save the expenses for disease surveillance.

Response: Thank you for your suggestion. While such networks can be generated based on many different network properties, we had a few properties for those two counties in hand. If such summary data for any other county are available, our method can be used with minimal or no modification at all to generate that network. The following was added in the discussion.

[Lines 226 - 228]

If additional data for movement networks in other locations become available, our network generation algorithms can be used with little or no modifications, depending on the data.

Comments on tables:

Table 1 – Abbreviations should be explained in the table legends.

Response: Thank you. We have updated the captions for this table to explain the abbreviations.

Table 1. Mixing matrix (probability of movement from row type to column type) for swine movement network [2]. The pig operation types are abbreviated as B (Boar Stud), F (Farrow), N (Nursery), G (Grower), and M (Market).

Figure 9 – How the percentages of vaccination were defined? A sensitivity analysis would be more profound than using these random numbers.

Response: Thank you for your suggestion. As different farms have different herd sizes (headcounts) it was difficult to compare results with respect to pigs vaccinated when we vaccinate the same number of farms in every strategy. Hence, we tried to match the actual amount (%) of pigs vaccinated by adjusting the number of farms for centrality based targeted vaccination schemes (As outlined in Table 6 of the 1st submission). However, in this new submission, we have added a new control measure (movement restrictions) along with hypothetical vaccination. We have also replaced old way of comparing the result based on percentages and replaced it with a new, easy to understand way of comparing control measure performance on a farm basis. We implemented control measures on a large range of values (no. of farms) in the current version. The new results are given in Fig 6 and 7.

---

## [Editor Report · Decision Letter 1]

23 Oct 2019

PONE-D-19-17842R1

Generation of swine movement network and analysis of efficient mitigation strategies for African swine fever virus

PLOS ONE

Dear Ferdousi,

Thank you for submitting your manuscript to PLOS ONE. After careful consideration, we feel that it has merit but does not fully meet PLOS ONE’s publication criteria as it currently stands. Therefore, we invite you to submit a revised version of the manuscript that addresses the points raised during the review process.

Could you please make some small edits:

line 52, a a......

Betwenness Betweenness

Delete all double spaces and replace by singles ones.

Standardize the use of capitals in titles of all articles in the reference list (use lowercase, see Plos One guidelines)

We would appreciate receiving your revised manuscript by Dec 07 2019 11:59PM. To enhance the reproducibility of your results, we recommend that if applicable you deposit your laboratory protocols in protocols.io, where a protocol can be assigned its own identifier (DOI) such that it can be cited independently in the future. For instructions see: http://journals.plos.org/plosone/s/submission-guidelines#loc-laboratory-protocols

We look forward to receiving your revised manuscript.

Kind regards,

Willem F. de Boer

Academic Editor

PLOS ONE

Additional Editor Comments (if provided):

Could you please make some small edits:

line 52, a a......

Betwenness Betweenness

Delete all double spaces and replace by singles ones.

Standardize the use of capitals in titles of all articles in the reference list (use lowercase, see Plos One guidelines)

---

## [Author Response · Author response to Decision Letter 1]

11 Nov 2019

Could you please make some small edits:

line 52, a a......

Response: Thank you for identifying the problem. We have fixed it.

Betwenness Betweenness

Response: Thank you. We have corrected it.

Delete all double spaces and replace by singles ones.

Response: Thank you for the suggestion. We searched for double spaces between words in the latex file. We found a few and fixed them.

Standardize the use of capitals in titles of all articles in the reference list (use lowercase, see Plos One guidelines)

Response: Thank you for the suggestion. We have fixed the unnecessary uppercases in the references. Please check the marked-up manuscript.

---

## [Editor Report · Decision Letter 2]

13 Nov 2019

Generation of swine movement network and analysis of efficient mitigation strategies for African swine fever virus

PONE-D-19-17842R2

Dear Dr. Ferdousi,

We are pleased to inform you that your manuscript has been judged scientifically suitable for publication and will be formally accepted for publication once it complies with all outstanding technical requirements.

With kind regards,

Willem F. de Boer

Academic Editor

PLOS ONE
---

## [Editor Report · Acceptance letter]

18 Nov 2019

PONE-D-19-17842R2 

Generation of swine movement network and analysis of efficient mitigation strategies for African swine fever virus 

Dear Dr. Ferdousi:

I am pleased to inform you that your manuscript has been deemed suitable for publication in PLOS ONE. Congratulations! Your manuscript is now with our production department. 

With kind regards,

on behalf of

Dr. Willem F. de Boer 

Academic Editor

PLOS ONE